# Variations in Yield, Essential Oil, and Salicylates of *Filipendula ulmaria* Inflorescences at Different Blooming Stages

**DOI:** 10.3390/plants12020300

**Published:** 2023-01-08

**Authors:** Kristina Ložienė, Jurga Būdienė, Urtė Vaitiekūnaitė, Izolda Pašakinskienė

**Affiliations:** 1Nature Research Centre, Institute of Botany, Žaliųjų Ežerų Str. 47, 08406 Vilnius, Lithuania; 2Nature Research Centre, Institute of Ecology, Akademijos Str. 2, 08412 Vilnius, Lithuania; 3Botanical Garden, Vilnius University, Kairėnų 43, 10239 Vilnius, Lithuania

**Keywords:** meadowsweet, natural habitats, inflorescences, volatile oil, salicylaldehyde, methylsalicylate

## Abstract

Meadowsweet (*Filipendula ulmaria*) is an essential oil-bearing, pharmacologically valuable medicinal plant growing wild in floodlands that are either not usually used for agriculture or have low economic value. The aim of this work was to understand the relationship between the yield of *F. ulmaria* inflorescences and the qualitative and quantitative composition of the essential oil during flowering stages in different habitats. Five different habitats of *F. ulmaria* were investigated for evaluation of inflorescence characteristics (length and weight) and the quantitative and qualitative composition of essential oils in early and late blooming stages (growing stage BBCH 62–63 and BBCH 65–67/72–73, respectively). The results showed significant (*p* < 0.05) positive correlations between the length and weight of inflorescences both in early and in late blooming stages (r = 0.73 and r = 0.72, respectively). The early blooming stage was observed to have 1.2–2 times greater quantity of volatile oils in comparison with the late blooming stage. Salicylaldehyde and methylsalicylate were the main salicylates in both blooming stages. Spearman’s correlation showed statistically significant relationship between percentages of salicylaldehyde and methylsalicylate (r = −0.94, *p* < 0.05). An increase in salicylaldehyde was accompanied by a decrease in methylsalicylate in essential oils. Statistically significant relations between the investigated parameters of plant and the parameters of habitat were not detected. The dried raw material yield of *F. ulmaria* in 1 ha in the late blooming stage was 18–56% higher than in the early blooming stage; however, the yield of essential oil in the early flowering stage was 1.5–1.6 times higher than in the late flowering stage. The obtained research data suggest that the low economic value of floodlands—the habitats of *F. ulmaria*—can be increased by using these areas as natural industrial plots.

## 1. Introduction

Meadowsweet *(Filipendula ulmaria* (L.) Maxim., syn. lat.: *Spiraeae ulmaria* (L.)) grows in well-hydrated areas and moist soils with full access to sun or in partially shaded places, marshy meadows, flooded meadows, alluvial meadows, riversides, lakesides, and ditches. This rhizomatous perennial species is common throughout the territory of the Baltic countries (Lithuania, Latvia, and Estonia), which make up a large part of the northeastern region of Europe on the eastern shores of the Baltic Sea and are located on the East European Plain. There are some large solid homogeneous “meadowsweet forests” (i.e., large areas) suitable for industrial collection of raw material of this medicinal plant in this region [1,2,3,4,5,6].

*F. ulmaria* essential oil-bearing species of the family Rosaceae growing wild in Europe and Western Asia is an ancient constituent of traditional medicine, the use of this plant dating back to the 12th century. Tinctures, dried-comminute herbal substances, and herbal teas for oral use are the most common types of meadowsweet pharmaceutical products applied in traditional or medical practices [7,8]. As an herbal substance, meadowsweet is known in traditional medicine for its wound-healing, gastroprotective, antimicrobial, antihyperalgesic, astringent, antiinflammatory, and other properties [9,10,11]. Meadowsweet as an herbal material is used not only for humans but also for animals. J.C. Harman (2007) explains that meadowsweet was used historically, and in modern veterinary applied as an antipyretic, antacid, and analgesic remedy, and also for treatment of bladder and kidney diseases and gastric ulcerations in horses [12]. According to an EMA assessment report made in 2011, *F. ulmaria* is available as a substance with marketing authorization in France and Spain, and is allowed as a healing product component, as well as as an ingredient in herbal teas, in Hungary, In the Czech Republic, it is only available as a food supplement (herbal tea), but besides the official marketing authorizations, meadowsweet has a wide range of traditional medicinal uses [7,8]. According to the latest findings, meadowsweet can also be used as an ingredient on the food market and in beverage production for its potential health-promoting effect as an antioxidant and antimicrobial (against *E. coli*, *E. faecalis*, and the fungi *P. cyclopium*, *F. oxysporum*) agent. Aqueous extracts of meadowsweet flowers were also applied in the experiment with pullulan films on prolonging apple stability in storage conditions, which also showed the applicability of antimicrobial and antifungal effects of meadowsweet in the food industry [13,14,15,16].

Salicylates (i.e., salicylic acid and its derivatives, such as methyl salicylate, salicylaldehyde, salicylalcohol and their glycosides) are well-known components of the *F. ulmaria* essential oil, comprising approximately 70% of its content; meadowsweet is one of the few plants that produce such a high percentage of salicylates [7,8,17,18]. Salicylates have a characteristic smell of bitter almond and are pharmacologically active substances with antiinflammatory, analgesic, antispasmodic, antibacterial effects and, in contrast to the synthetic analogue with the trade name aspirin, do not have a negative effect on the digestive system [14,19]. Salicylates are also used in the cosmetic industry as anti-aging, anti-inflammatory, or anti-oil skin substances, and as flavouring substances in drinks, foods, and tobacco products [11,15,20,21,22,23].

According to the European Pharmacopoeia (2020) [24], the raw plant material of meadowsweet is made of its dried flowering tops (inflorescences, *Filipendulae ulmariae herba*) with a minimum of 1 mL/kg of essential oil (dried drug). Meadowsweet usually has white–yellowish blossoms arranged in dense irregular panicle-form inflorescences with elongated lateral branches and blooms from June to August. The long flowering period and irregular panicle-form inflorescences complicate the collection of raw material at the full-bloom period: inflorescences have fully flowering twigs, not-yet-flowering twigs, and twigs with fruit at the same time. Previous studies showed that methyl salicylate and salicylaldehyde amounted to 28.2% and 2.8%, respectively, in *F. ulmaria* essential oil during the flowering stage; however, during the fruiting stage, the percentage of salicylaldehyde decreased (to 12.4%) and methyl salicylate increased (up to 11.2%) [25]. Therefore, the flowering period can influence the yield and composition of *F. ulmaria* essential oil.

The aim of this work was to evaluate the relationship between the yield of *F. ulmaria* inflorescences and the qualitative and quantitative composition of the essential oil during the flowering stage in different habitats.

## 2. Results

### 2.1. Habitat Characteristics

Descriptions of the habitats are presented in Table 1. The Biržai and Pabiržė habitats were located in Mūša-Nevėžis climatic subdistrict of the central district of Lithuania, Kernavė, Vilnius, and Mikniškės habitats in the Dzūkija climatic subdistrict of the southeastern Lithuanian highland district (Figure 1). All habitats were located on a flat or a slightly rolling relief and had very good lighting. Two habitats (Biržai and Vilnius) belonged to *Calthion-palustris* and three habitats (Pabiržė, Kernavė, and Mikniškės) belonged to *Alopecurion pratensis* vegetation alliances. All the habitats had a high total herb covering even though the cover-abundance of 30–52% of the species in the habitats scored a value of 2 according to the Braun–Blanquet scale [26]. *F. ulmaria* cover-abundance was the highest in all the habitats and varied from a value of 3 to a value of 5 (Table 1).

The analysis of soil showed that the soil pH of *F. ulmaria* habitats varied from acid to neutral (Table 2). Mobile phosphorus, total magnesium, and organic carbon varied very widely; the minimum and maximum values differed 7, 10, and 11 times, respectively. The soil in Vilnius habitat had the highest acidity (pH value 5.7) and the highest amount of organic carbon (28.9%). On the other hand, the soil in the Biržai habitat was the most alkaline (pH value 7.3) and had the highest amount of magnesium (3312 mg/kg). The Mūša-Nevėžis and Dzūkija climatic subdistricts differed significantly (*p* < 0.05) with regard to the amount of mobile potassium in the soil.

The average number of inflorescences in a habitat was 47.5 ± 20.4 inflorescences per m^2^. The Vilnius and Biržai habitats showed the highest number of *F. ulmaria* inflorescences per area unit. In these two habitats, the number of inflorescences per m^2^ was twice as high compared to other habitats (Table 1). Spearman’s correlation showed a negative relationship of inflorescence number/m^2^ to pH, mobile phosphorus and potassium, but positive with regard to total magnesium and organic carbon; however, these relationships between inflorescence number/m^2^ and all the investigated edaphic characters were not statistically significant.

### 2.2. Soil Chemistry and Its Effect on Composition of Filipendula ulmarias Essential Oils

The average length and weight of *F. ulmaria* inflorescences was 12.0 ± 4.1 cm and 3.8 ± 2.7 g, respectively, in the early blooming stage (N = 500) and 14.5 ± 4.9 cm and 4.3 ± 2.6 g in the late blooming stage (N = 500). The inflorescences are elongated by 21% and weighted by 13% during flowering. The positive correlation was established between the length and weight of the inflorescences both in the early blooming stage (r = 0.73, *p* < 0.05) and in the late blooming stage (r = 0.72, *p* < 0.05). The *t*-test showed statistically significant differences between inflorescence length in the early blooming stage and inflorescence length in the late blooming stage (*p* < 0.05); meanwhile, no significant differences between these blooming stages were detected with regard to inflorescence weight.

Kernavė had visibly higher values in inflorescence length and weight in both the early and late blooming periods in comparison to other habitats. The lightest and shortest inflorescences in both blooming stages were established in Pabiržė (Table 3). Inflorescences lengthened more, then became heavier during the flowering period in all investigated habitats (except the Pabiržė habitat). It is interesting to note that the inflorescences did not grow in weight and even lost up to 15% weight in the Mikniškės and Vilnius habitats, respectively (Table 3). No significant correlation between the investigated parameters of inflorescences and the soil chemistry of the habitats was detected, and no significant differences between both climatic subdistricts with regard to inflorescence parameters were detected either.

### 2.3. Chemical Composition of Essential Oils

The early blooming stage was observed to have 1.2–2 times greater quantity of volatile oils in comparison with the late blooming stage: essential oil percentage was 1.3 ± 0.33% and 0.61 ± 0.35% in *F. ulmaria* inflorescences during early and late blooming, respectively (Figure 2). A statistically significant (*p* < 0.05) distinction in the percentage of essential oil between different blooming stages was observed in all investigated habitats. The variation of essential oil percentage in the early blooming stage among the habitats was higher than the variation in the late blooming stage (CV = 25 and CV = 9, respectively). The highest percentage of volatile oil was observed in Biržai, where it reached 1.66 ± 0.018% in the early blooming stage (Figure 2). The opposite situation was detected in Kernavė, where the percentage of essential oil in the early blooming stage was the lowest and very similar to the percentage established in the late blooming stage (the difference was 0.13% only); however, this difference, as was mentioned above, was significant. Although the majority of investigated habitats significantly (*p* < 0.05) differed from each other (except for the Pabiržė and Mikniškės habitats), according to essential oil percentage in the early blooming stage, they did not statistically differ by this indicator at the late blooming stage (Figure 2). No statistically significant relation between the percentage of essential oil, parameters of the inflorescences and soil chemistry was detected; the climatic subdistricts did not differ statistically with regard to essential oil percentage in the early and late blooming stages.

Salicylates (salicylaldehyde, methyl salicylate, ethyl salicylate, benzyl salicylate) constituted the highest portion of essential oils (from 64.21% to 94.83% in Biržai and Pabiržė habitats, respectively) in both blooming stages (Appendix A). Salicylaldehyde (72.30 ± 4.71% and 65.66 ± 20.94% in early and late blooming stage, respectively) and methyl salicylate (18.41 ± 3.29% and 17.10 ± 10.72% in early and late blooming stage, respectively) were not only the main salicylates, but also the most common substances of the essential oils; meanwhile, ethyl salicylate and benzyl salicylate were very rarely detected chemical compounds, or were not found at all. Salicylaldehyde prevailed over methyl salicylate in all the essential oils and, depending on the habitat and the blooming stage, the rate of these salicylates varied from 6:1 to 2:1. Spearman’s correlation showed a statistically significant relationship between percentages of salicylaldehyde and methyl salicylate (r = −0.94, *p* < 0.05). The increase in salicylaldehyde was accompanied by the decrease in methyl salicylate in the essential oils. The variation of salicylaldehyde percentage between the habitats in both the early and late blooming stages was lower (CV = 7% and CV = 32%, respectively) than the variation of methyl salicylate percentage (CV = 18% and CV = 63%, respectively); meanwhile, the variation of the percentage sum of these two salicylates was even lower: the value variation coefficient was 3% in the early and 14% in the late blooming stage. The highest percentage of salicylaldehyde was detected in the essential oil isolated from late blooming inflorescences collected in Biržai: salicylaldehyde amounted to 94.81% of essential oil; meanwhile, methyl salicylate was found in traces only. By comparison, the lowest percentage of salicylaldehyde was detected in late blooming inflorescences collected in Pabiržė. The significant correlation was detected between methyl salicylate percentage and salicylaldehyde percentage in the late blooming stage (r = 0.9, *p* < 0.05), and between salicylaldehyde percentage in the late blooming and inflorescence length in the early blooming stage (r = 0.9, *p* < 0.05). No statistically significant correlations were detected between salicylaldehyde percentage and soil chemistry or between methyl salicylate percentage and the chemical parameters of the soil, and no significant difference between both climatic subdistricts with regard to percentage of salicylaldehyde and methyl salicylate was detected either.

### 2.4. The Calculated Yield of Dried Raw Material and Essential Oil

Table 4 demonstrates the calculated values of *F. ulmaria* dried raw material yield in ha and the yield of essential oil isolated from this raw material. The most productive habitats with regard to dried raw material and essential oil yields were in Biržai and Vilnius. *F. ulmaria* dried raw material yield in ha was 18–56% higher in the late blooming stage than in the early blooming stage (except for the Vilnius habitat). The essential oil yield in the early flowering stage was often 1.5–1.6 times higher than in the late flowering stage.

## 3. Discussion

About two-thirds of different medical plants in use are obtained from natural habitats and only one-tenth of commercial medical herbs are grown in Europe [27]. That is why it is so important to examine the factors that could affect the quality of collected raw material. The quality is usually associated with all the biologically active substances in a medical plant, such as terpenes, polyphenols, flavonoids, carotenoids, vitamins, saponins, etc. Perennial herbaceous plants of two species of the genus *Filipendula*, growing wild throughout the Baltic region, occur in different habitats. *F. vulgaris* plants (dropwort, called broad-flower and commonly known as fern-leaf dropwort) grow in lime-rich meadows; meanwhile plants of *F. ulmaria* (commonly known as meadowsweet or mead wort) grow in floodlands [3,28]. Therefore, meadows with *F. ulmaria* are not usually used for agriculture. The plant communities of the investigated *F. ulmaria* habitats belonged to the associations of two alliances of phytocoenological vegetation classes: *Molinio-Arrhenatheretea elatioris* R. Tx. 1937: *Calthion palustris* R. Tx. 1937 em. Lebrun et al. 1949 and *Alopecurion pratensis* Passarge 1964 (Table 1). The meadows of these alliances (especially of *Calthion palustris* alliance) are of low economic value [5]. However, these associations can be used as a source of *F. ulmaria* pharmacological raw material. *F. ulmaria* cover-abundance 3, 4, and 5 (Table 1) means that 1/4–1/2, 1/2–3/4 and more than 3/4 of the habitat, respectively, is covered with plants of this species. Such large cover-abundance enables them to collect large amounts of pharmacological raw materials of *F. ulmaria*— their blooming tops.

It is known that not only genetic factors affect the morphological and chemical parameters of plants but that the contribution of species-friendly environmental conditions (edaphic, climatic, etc.) can also be significant for these parameters, leading to the abundance of species richness in habitats and the essential oil accumulation [29,30]. During late blooming, when a large portion of inflorescences have ripening fruits (there are no fruits during early flowering), plants spend more energy on fruit ripening than on flowering. Therefore, different environmental factors throughout habitats can have higher influence on essential oil in the early blooming stage. However, even though there were considerable differences of soil pH, the amount of mobile phosphorus, mobile potassium, total magnesium and organic carbon in the examined *F. ulmaria* habitats, no significant relation between the morphological and chemical parameters of *F. ulmaria* inflorescences and the soil chemistry of the habitats was detected either in the early or late blooming stage. Two *F. ulmaria* habitats were located in the central district of Lithuania (in the Mūša-Nevėžis climatic subdistrict), and three habitats were located in the southeastern Lithuanian highland district (in the Dzūkija climatic subdistrict) (Figure 1). These climatic districts (and also subdistricts) differ in average annual temperature, the amount of precipitation per year, and the duration of snow preservation. The Mūša-Nevėžis climatic subdistrict has the lowest annual rainfall (500–620 mm) in Lithuania [31,32]. However, the statistical analysis showed that the habitats of the central Lithuania district did not significantly differ from the habitats of the southeastern Lithuanian highland district with regard to essential oil percentage in both the early blooming stage (0.77 ± 0.09% and 0.58 ± 0.17%, respectively) and late blooming stage (0.02 ± 0.03% and 0.14 ± 0.17%, respectively). Moreover, with regard to parameters and percentages of salicylaldehyde and methyl salicylate in the essential oil of *F. ulmaria* inflorescences.

*F. ulmaria* blooms for 2–3 weeks (depending on weather conditions) from the beginning of July. Such long blooming could prolong the inflorescence collection time; however, the raw material collected at late blooming is partially with fruits. The results of the research have shown that the blooming stage is a more relevant factor to describe essential oil production than the criteria of length or weight for meadowsweet inflorescences. The percentage of the essential oil obtained from early blooming *F. ulmaria* inflorescences (1.31 ± 0.33%) was higher than stated in the quality requirements for this raw material in the European Pharmacopoeia (the minimum 1 mL/kg) and EMA assessment (also, the minimum 1 mL/kg) [7,8,24]. The late blooming stage appeared to be less productive than the EMA or the European Pharmacopoeia suggests (the mean of essential oil yield in the late blooming stage was 0.61 ± 0.35%). A wider contrast gap appears when comparing these two sources and the current research data to the study of M. G. Valle: the flower material of 25–30 cm length, collected in Italy, only reached the amount productivity of 0.04% in essential oil yield [33]. Even though the present results have not shown any statistically significant relation between the percentage of essential oil and inflorescence length, a very low percentage of essential oil established in M. G. Valle’s study may be related to the length of collected blooming tops. They could have been longer, in comparison with the present study data, and included leaves that do not accumulate essential oil. It may also be related to the less favorable climatic conditions for *F. ulmaria* in Italy, as well as to the type of apparatus (not Clevenger-type apparatus) used for the extraction of essential oil.

In terms of salicylates, meadowsweet is one of the few herbs (willow (*Salix* L.); wintergreen (*Gaultheria procumbens*) and poplar (*Poplar* L.)) that are rich in these compounds. Although the percentage of salicylates in the essential oil was in most cases lower in the late blooming period, a higher percentage of other constituents was observed. In the range of different *Filipendula* species, the chemical composition might be considered rather similar. According to the study carried out by M. Pavlovic et al., in *Filipendula hexapetala* Gilib., the quantity of salicylates reached 27.8% of the total percentage of its chemical composition [34]. The data about other *Filipendula* species published by N. Radulović et al., in which *F. vulgaris* Moench was tested for the number of salicylates mentioned in this work, showed even more similarity to our research, with the result that 71% of essential oil had salicylates [35]. The mentioned works show visible similarities, like the ones that all of the studied *Filipendula* species produce salicylates that are part of their essential oils, as well as some other similar chemical compounds (linalool, hexadecanal, tricosane, and etc.), although not all of the species have such high amounts of salicylates [34,35]. Salicylaldehyde as a major compound exceeding more than half of the essential oil content, and methyl salicylate as the second high was found in essential oils of *Filipendula vestita* leaves from the western Himalaya [36]. Despite quite similar results to ours with regard to the essential oil composition, according to the presented data, the essential oil yield from leaves was 0.3% (*v/w*). The data on variation in salicylate composition in the early and late blooming stages is scarce. By analysing the obtained results, we observed that the increase in salicylaldehyde in the late blooming stage was accompanied by the decrease in methyl salicylate, and vice versa. Two other salicylates (ethyl and benzyl) were found in very low amounts, so the changes in their quantities do not depend on the blooming stage. K. Stawaczyk and his team studied a relationship between the content of salicylates in meadowsweet extracts and their antioxidant properties [37]. They concluded that the variation of phenolic compounds, including salicylates, mostly depends on geographical distances between the samples and ecological characteristics of the area. In our research, we observed that a geographical position of *F. ulmaria* in the habitats of Lithuania does not affect the composition of the inflorescence essential oil as significantly as the blooming stage.

## 4. Materials and Methods

### 4.1. Habitats and Plant Material

Five phytocenologically homogeneous habitats of *F. ulmaria* were investigated from June until August in 2018 in Lithuania (the Baltic country in the northeastern region of Europe). *F. ulmaria* habitats were chosen randomly, distances among habitats were no less than 10 km (maximum distances among habitats in the northern and southeastern part of Lithuania were 150–250 km). Study sites are presented in Table 1 and Figure 1. The phytocoenology study was made in 16 m^2^ fields of meadows according to Braun–Blanquet methodology [26]. Plant communities were distinguished according to the vegetation classification systems proposed by J. Balevičienė (1991) and J. Balevičienė et al. (1998) [5,38]. The relative lightening of the habitats has been assessed visually. Attribution of *F. ulmaria* habitats to climatic districts and subdistricts of Lithuania was carried out on the basis of Lithuanian climatic zoning according to K. Kaušyla [31].

Samples of soil were collected from each habitat separately and dried at room temperature. Each sample of topsoil was prepared in the following way: 5–7 subsamples (subject to the area of the habitat; each subsample ~100 g) were taken from the depth of 10–15 cm (plant rhizosphere) by using the envelope principle, with the distance of 1 m from the central point of the habitat, and homogenised. The content of humus and mobile phosphorus (P_2_O_5_) was estimated photoelectrocolorimetrically. The content of mobile potassium (K_2_O) was estimated photometrically by flame photometry and magnesium—. The soil pH was estimated electrometrically by using 1 M KCl solution. The soil analysis was done at the Agrochemical Research Laboratory of the Lithuanian Research Centre for Agriculture and Forestry (Kaunas, Lithuania).

In this study, only the inflorescences were investigated and used as a primary herb harvest. The plant material in each habitat was collected separately in growth stages BBCH 62–63 (the early blooming stage in the text) and growth stage BBCH 65–67/72–73 (the late blooming stage in the text), in June and in August, respectively [39]. The interval between these two collections was 19–21 days. The calculations of fresh inflorescences of *F. ulmaria* have been assessed by measuring 100 random inflorescences in each population separately. Samples of raw material (mixes of 100 random) inflorescences in each habitat) were collected in such a way that they could represent the whole habitat, i.e., the collection of inflorescences was carried out randomly across the whole habitat. Thus, a total of 10 samples of raw material from five different habitats were collected for further research. Each fresh inflorescence was weighed, measured in length from the beginning of the panicle up to the end of the longest lateral branch and dried at room temperature for further analysis. After drying, the same 100 measured inflorescences of each population were weighted repeatedly (constant mass of air-dried material for seven days) to evaluate the weight loss due to drying.

### 4.2. Isolation of Essential Oil

The hydrodistillation of essential oils from dry inflorescence (stems removed) samples from each habitat was carried out separately by the Clevenger-type apparatus for 2 h [24]. The yield of essential oil was calculated in % (*v/w*), based on dry plant weight. The distillation of essential oil from each sample was carried out in three repetitions (the only sample collected in the Pabiržė habitat in the late blooming stage was distilled in the one repetition due to the low amount of raw material). The isolated essential oils were stored in the freezer until further gas chromatography-mass spectrometry (GC-MS) analysis.

### 4.3. Essential Oil Analysis by GC-MS

GC-MS was applied to different essential oils from both blooming stages (early and late blooming) of all five population locations. The essential oils from the early blooming stage and essential oils from the late blooming stage were subjected to 10 separate GC-MS analyses in total. A total of 10 µL of each essential oil were dissolved in 1 mL of pentane and diethyl ether (1:1) mixture. A total of 1 µL of prepared solution was injected into GC/MS system. Analyses were performed by using Shimadzu GC/MS-Q2010 PLUS interfaced to Shimadzu GC-MS-QP2010 ULTRA mass spectrometer and fitted with a Rxi-5MS (30 m × 0.25 mm × 0.25 µm) capillary column. Mass spectra in electron impact mode were generated at 70 eV, 0.97 scans per second, mass range 33–400 m/z. The oven temperature of the gas chromatograph was set at 50 °C (for 1 min) and then increased by 5 °C per minute to reach 160 °C, then held for 2 min and programmed to reach 250 °C at the increased rate of 10 °C/min, and hold at the final temperature for 4 min. Helium (1.0 mL/min), at ratio 1:10, was used as carrier gas. Detector and injector temperatures were 250 °C, ions source temperature was 220 °C. The qualitative analysis and identification of compounds was based on the comparison of time and retention indexes of the column, with corresponding data in literature [40], as well as computer libraries of mass spectra (using “GC/MS solution” v. 2.71 software from Shimadzu and Wiley and NIST). The identification of the compound was approved if mass spectra library data matched the computer data with probability equal to 90% or above. The retention indices were determined with regard to retention times of a series of *n*-alkanes (C_7_–C_30_) with linear interpolation. The relative percentage of essential oil constituents was computed from the chromatogram peak areas with none of the correction factors.

### 4.4. Statistical Data Analysis

Statistical data processing, including the calculation of means, standard deviations, Spearman’s rank correlation coefficients (r), the *t*-test, the Kruskal–Wallis and reliabilities (p), was carried out with the STATISTICA^®^ 7.0 and MS Excel 2016. The variation coefficients were used to evaluate variations of morphological traits and chemical compounds of inflorescences in early and late blooming stages. The visualization of the data was obtained by Microsoft Excel program.

## 5. Conclusions

Plants of *F. ulmaria* grow naturally in a floodlands that are not usually used for agriculture or have low economic value. However, *F. ulmaria*, as a pharmacologically valuable species, can increase the economic value of such meadows and, as a result of it, they could be used as natural industrial plots. The results have shown that salicylates comprised the greater part of essential oils in inflorescences in both blooming, that the opportunities for collecting raw material (flowering tops) in natural habitats are great and that they increase in the later flowering stage of the plants. However, even though the yield of dried raw material is higher in the late flowering stage, its collection in the early flowering stage is more valuable in terms of essential oil content, because the essential oil yield at that time can amount up to 49–64% in comparison to the essential oil yield in late flowering stage in the same area.

## Figures and Tables

**Figure 1 plants-12-00300-f001:**
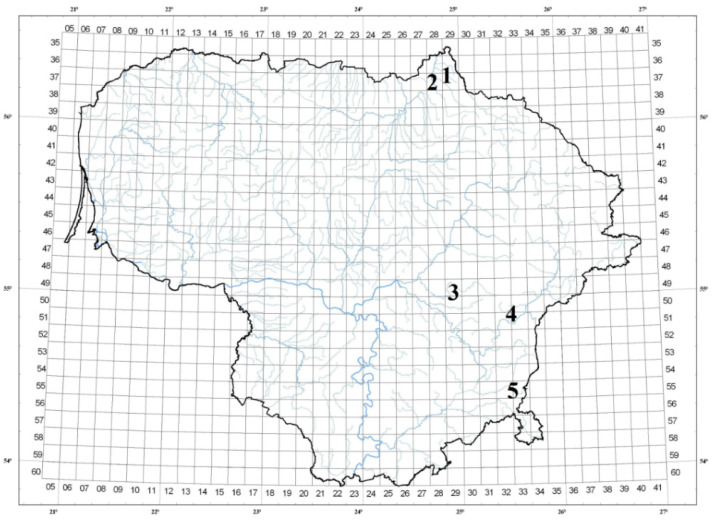
Map of study field distribution in different parts of Lithuania. Yield collection sites allocated in Biržai (1), Pabiržė (2), Kernavė (3), Vilnius (4), and Mikniškės (5).

**Figure 2 plants-12-00300-f002:**
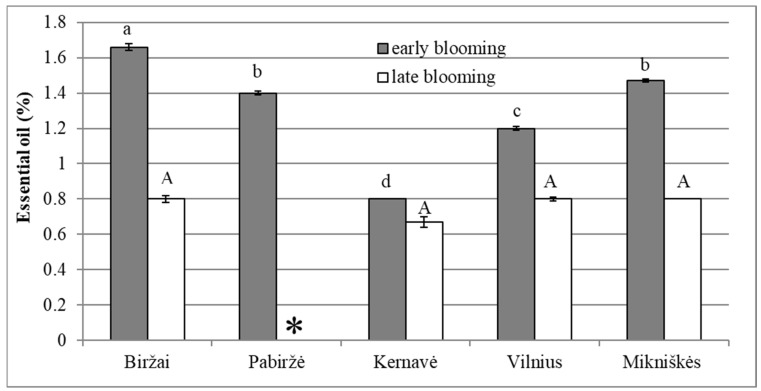
Comparison of essential oil yields of *Filipendula ulmaria* inflorescences harvested in various habitats in early and late blooming stages. Letters denote statistically significant (*p* < 0.05) differences between different habitats. Lower-case letters represent the early blooming stage, and capital letters represent the late blooming stage (* low amount of essential oil did not allow to estimate its percentage in sample).

**Table 1 plants-12-00300-t001:** General characteristics of *Filipendula ulmaria* (L.) Maxim. investigated habitats (Cp—*Calthion palustris* R. Tx. 1937 em. Lebrun et al. 1949, Ap—*Alopecurion pratensis* Passarge 1964, SD—standard deviation).

No.	Location	Coordinates (WGS-84)	Area (ha)	Vegetation Alliance	Relief	Lightning (%)	*F. ulmaria* Cover-Abundance *	Number of Inflorescences in m^2^ (mean ± SD)
1	Biržai	E24.763386 N56.183361	1.5	Cp	plane	100	4	66 ± 17
2	Pabiržė	E24.634662 N56.183625	1.2	Ap	plane	100	4	34 ± 8
3	Kernavė	E24.850016 N54.877099	0.9	Ap	plane	100	4	32 ± 12
4	Vilnius	E25.303289 N54.769149	2.1	Cp	plane	100	5	73 ± 24
5	Mikniškės	E25.517729 N54.449071	0.1	Ap	wave	90	3	33 ± 10

*According to Braun–Blanquet scale [26].

**Table 2 plants-12-00300-t002:** Chemical parameters of the soil in different *F. ulmaria* habitats.

No.	Location	Chemical Parameters				
		pH	Mobile Phosphorus (mg/kg)	Mobile Potassium (mg/kg)	Total Magnesium (mg/kg)	Organic Carbon (%)
1	Biržai	7.3	111	108	3312	5.5
2	Pabiržė	6.6	147	122	1116	8.3
3	Kernavė	6.9	240	82	570	6.4
4	Vilnius	5.7	36	73	540	28.9
5	Mikniškės	6.5	64	77	318	2.6

**Table 3 plants-12-00300-t003:** Lengths and weights of *Filipendula ulmaria* fresh inflorescences (N = 100 in each habitat) collected in early and late blooming stages, in different habitats (SD, standard deviation, CV, coefficient of variation).

Parameters of Fresh Inflorescences			Habitats				
			Biržai	Pabiržė	Kernavė	Vilnius	Mikniškės
Early blooming	Length (cm)	Average ± SD	12.7 ± 3.1	10.1 ± 3.3	14.3 ± 5.5	11.4 ± 3.9	11.6 ± 3.1
		Min	5.3	4.5	5.8	3.0	4.2
		Max	22.2	20.1	31.4	26.0	19.5
		CV (%)	24	33	39	34	27
	Weight (g)	Average ± SD	3.9 ± 2.3	2.9 ± 1.9	5.0 ± 4.1	4.1 ± 2.3	4.2 ± 2.0
		Min	0.9	0.5	0.9	1.0	0.9
		Max	12.1	8.3	27.5	13.6	10.4
		CV (%)	59	66	82	56	48
Late blooming	Length (cm)	Average ± SD	14.0 ± 3.7	12.4 ± 4.4	18.8 ± 5.3	14.1 ± 3.3	13.4 ± 4.5
		Min	5.3	3.5	7.6	6.8	3.0
		Max	24.7	24.4	36.4	23.9	28.3
		CV (%)	26	33	28	23	34
	Weight (g)	Average ± SD	4.2 ± 2.5	3.8 ± 2.6	5.8 ± 3.1	3.5 ± 1.3	4.2 ± 2.5
		Min	0.51	0.4	1.2	0.6	0.5
		Max	19.0	15.2	14.0	7.5	15.5
		CV (%)	60	68	53	37	60

**Table 4 plants-12-00300-t004:** Calculated yield of *Filipendula ulmaria* dried raw material and essential oil in different habitats (*low amount of essential oil did not allow to estimate its percentage in habitat).

Yield	Blooming Stage	Habitat				
		Biržai	Pabiržė	Kernavė	Vilnius	Mikniškės
Yield of dried raw material (inflorescences) (kg/ha)	Early	850	320	530	990	450
	Late	1080	500	720	1000	530
Yield of essential oil (l/ha)	Early	14.1	4.5	4.2	11.9	6.6
	Late	8.6	*	4.8	8	4.2

## Data Availability

Not applicable.

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
