# Peer review of "Variations in Yield, Essential Oil, and Salicylates of Filipendula ulmaria Inflorescences at Different Blooming Stages"

_plants, 2023, doi:10.3390/plants12020300_

Round 1
Reviewer 1 Report
The aim of the work is to compare qualitative and quantitative composition of essential oil obtained from F. ulmaria purchased during flowering stages in different habitats. The issue of the studies is interesting. Authors decided to characterize plant which reveal valuable pharmacological activities. On the whole, paper is well presented along with detail description of the plant, experimental procedure, obtained results and discussion. Statistical analysis is provided. The research goal was achieved. Please see the following comments:
Abstract: informative but too long. It should be reduced
Introduction: informative and readable. References are provided
Results: well presented in form of tables and figures. Authors presented the most important study results along with comments and their detailed analysis. Please improve the point:
1. Whole Table 1 should include statistic information as in the column ‘Number of inflorescences in m2’
Discussion: well described along with u-to-date references. Authors discussed al obtained study results and compared them with results presented in other papers.
Materials and methods: isolation and GC-MS analysis should be separated. Why Authors decided to use hydro-distillation? Are there works that present this type as the best? Please provide the papers as references. If there are no such works, it would be valuable to carry out several types of distillation and choose the best one.
Conclusions: well prepared
References: up to date and adequate
Author Response
Abstract: informative but too long. It should be reduced
Author’s response. Yes, we reduced it.
Introduction: informative and readable. References are provided
Results: well presented in form of tables and figures. Authors presented the most important study results along with comments and their detailed analysis. Please improve the point:
- Whole Table 1 should include statistic information as in the column ‘Number of inflorescences in m2’
Author’s response. Yes, we included statistical information.
Discussion: well described along with u-to-date references. Authors discussed all obtained study results and compared them with results presented in other papers.
Materials and methods: isolation and GC-MS analysis should be separated.
Author’s response. In the Materials and Methods section, the essential oil isolation and GC-MS analysis parameters were separated into two sections.
Why Authors decided to use hydro-distillation? Are there works that present this type as the best? Please provide the papers as references. If there are no such works, it would be valuable to carry out several types of distillation and choose the best one.
Author’s response. “Essential oil, also defined as essence, volatile oil, etheric oil or aetheroleum, is a complex mixture of volatile constituents biosynthesized by living organisms. Essential oils can be liberated from their matrix by water, steam and dry distillation, or expression in the case of citrus fruits.” (H. K. BaÅŸer, 2007). Cold pressing is used to extract citrus EO, steam distillation and dry distillation are more industrially used methods, and hydro-distillation is recommended by the European Pharmacopoeia. This is why hydro-distiliation was chosen for the extraction of essential oil.
Hüsnü, K., BaÅŸer, C., Demirci, F. (2007). Chemistry of Essential Oils. In: Berger, R.G. (eds) Flavours and Fragrances. Springer, Berlin, Heidelberg. https://doi.org/10.1007/978-3-540-49339-6_4.
Conclusions: well prepared
References: up to date and adequate
Reviewer 2 Report
Dear authors! Thanks for interesting work!
Please find some comments for article improvement.
The aim of research should be more concentrated and summarized - not for 4 lines. Is it Ok to formulate with term – to understand?
Line 16 - Please add correlation coefficient
Lines from 81-87 – it should be at the beginning of introduction
Lines 92 – in the aim is mentioned Baltic states but in the aim presented in abstract - not.
In all article – please choose one abbreviation of samples – in same cases is written No.1., No.2. , in other places Birži etc. Please remove these numbers for example in Lines 99 and 100. In map it is OK with numbers.
Presentation of Table 1. should be improved – it is too big and in next page also head of table should be presented. In some rows just one letter - for example letter C. Some of information could be added in other table in materials and methods – for example coordinates. Or may be you can change direction of text in head of table un use some abbrev – for example, plane - P etc. Table should be improved to be more clear.
Please check if there are consistency for data in Lines 24-27 and Lines 123-126, and lines 146-149.
Please remove chart are borders for figures
In Figure 2 are presented the same data as in Table 2 (for fresh plants). Data should be presented once. I suggest to add Box plots showing min, max etc.
In figures SD and statistical analyses should be added showing factors significance and differences between samples.
197 and 198 lines should be improved – how is possible to calculate relationship with soil chemistry – there should be certain parameters.
Also data presented in Figure 4 are in Table 3 – they should be presented once.
Table with a lot of compounds in small percentage - is it necessary to present them all. May be add this table in supplementary data.
Composition could be presented based data analyses – PCA, cluster with heatmap – it will be more representative for readers.
How is SD for data in Table 3? Please add.
Line 254 – why t-test is used – there are comparison of more than two samples.
Please correct Lines 307-309 – you have not data of Baltic states.
Line 335 – why two times of analyses where selected, not in full blooming. Could be there some differences?
Line343 – how long they were dried – up to certain moisture, certain duration – did you have the same moisture for all of them when yield of dried herbs were analysed?
Line 350 – experiments were carried out in 1-3 repetitions? Please clarify how it is possible? Some samples was analysed just once?
Line 373 – why t-test only?
In conclusions – please add something about essential oil chemical components
References should be undated with some more recent ones- - latest is from 2018, but a lot of them are much older.
Author Response
The aim of research should be more concentrated and summarized - not for 4 lines. Is it Ok to formulate with term – to understand?
Author’s response. Yes, we have shortened the aim of the research. “To understand” was changed to “to evaluate”.
Line 16 - Please add correlation coefficient
Author’s response. Correlation coefficients were added.
Lines from 81-87 – it should be at the beginning of introduction
Author’s response. The text of these lines was transferred to the beginning of the introduction.
Lines 92 – in the aim is mentioned Baltic states but in the aim presented in abstract - not.
Author’s response. We shortened the aim of the research in the Introduction excluding the mention of the Baltic states. We did not mention the Baltic states in the abstract either.
In all article – please choose one abbreviation of samples – in same cases is written No.1., No.2., in other places Birži etc. Please remove these numbers for example in Lines 99 and 100. In map it is OK with numbers.
Author’s response. Yes, we corrected this.
Presentation of Table 1. should be improved – it is too big and in next page also head of table should be presented. In some rows just one letter - for example letter C. Some of information could be added in other table in materials and methods – for example coordinates. Or may be you can change direction of text in head of table un use some abbrev – for example, plane - P etc. Table should be improved to be more clear.
Author’s response. Yes, we improved Table 1: the soil parameters of the habitats were transferred to a separate new Table 2. Thus, Table 1 became shorter and clearer.
Please check if there are consistency for data in Lines 24-27 and Lines 123-126, and lines 146-149.
Author’s response. Yes, we adjusted the consistency of the data in the text of these Lines.
Please remove chart are borders for figures
Author’s response. Yes, we have removed it.
In Figure 2 are presented the same data as in Table 2 (for fresh plants). Data should be presented once. I suggest to add Box plots showing min, max etc.
Author’s response. Yes, in Figure 2, were presented the same data as in Table 2. Therefore, Figure 2 and the sentence, where this Figure 2 was cited, was deleted.
In figures SD and statistical analyses should be added showing factors significance and differences between samples.
Author’s response. Yes, in Figure 2 we added SD and statistical analyses showing the factors’ significance between the habitats.
197 and 198 lines should be improved – how is possible to calculate relationship with soil chemistry – there should be certain parameters.
Author’s response. We checked it again: no significant correlation between the investigated parameters of inflorescences and the soil chemistry of the habitats was determined.
Also data presented in Figure 4 are in Table 3 – they should be presented once.
Author’s response. Yes, we deleted Figure 4, because the data presented in this figure are also in Table 3 (this table is in the Supplement now).
Table with a lot of compounds in small percentage - is it necessary to present them all. May be add this table in supplementary data.
Author’s response. Yes, we added Table 3 in the supplementary data.
Composition could be presented based data analyses – PCA, cluster with heatmap – it will be more representative for readers.
Author’s response. The aim of this work was to understand the relationship between the yield of F. ulmaria inflorescences and the qualitative and quantitative composition of the essential oil during flowering stages but not to investigate the diversity of F. ulmaria. Therefore, we studied only five habitats. Five habitats are too few to apply PCA or cluster with heatmap statistical analyses.
How is SD for data in Table 3? Please add.
Author’s response. Despite the fact that the hydro-distillation of each sample was carried out in triplicate, to determine the exact yield, the essential oils were pooled together, and their chemical composition analyses were carried out only once. As a result, in the table (Supplementary material Table) with all the identified compounds, the standard deviations of the compound levels are not given.
Line 254 – why t-test is used – there are comparison of more than two samples.
Author’s response. Yes, we corrected this.
Please correct Lines 307-309 – you have not data of Baltic states.
Author’s response. Yes, we corrected this.
Line 335 – why two times of analyses where selected, not in full blooming. Could be there some differences?
Author’s response. Meadowsweet usually has white-yellowish blossoms arranged in dense irregular panicle-form inflorescences with elongated lateral branches, it blooms from June to August. The long flowering period and irregular panicle-form inflorescences complicate the collection of raw material during the full-bloom period: inflorescences have fully flowering twigs, not-yet-flowering twigs and twigs with fruit at the same time. Therefore, two times of analyses were selected – early blooming and late blooming. All this explanation is provided in the fourth paragraph of the Introduction.
Line343 – how long they were dried – up to certain moisture, certain duration – did you have the same moisture for all of them when yield of dried herbs were analysed?
Author’s response. Before extraction of the essential oils by hydro-distillation, the plant material was dried at room temperature, without sunlight, until the mass of the air-dried material was constant at three decimal places for seven days. A constant mass of plant material for seven days confirms the maximum loss of moisture in the sample. All samples were prepared in the same way. Drying at temperatures above room temperature or with blowing would likely increase the evaporation of moisture, but at the same time, there is a possibility of losing small volatile organic molecules that are components of the essential oil.
.
Line 350 – experiments were carried out in 1-3 repetitions? Please clarify how it is possible? Some samples was analysed just once?
Author’s response. We carried out the experiment in 3 repetitions. Only the sample collected in Pabiržė habitat in the late blooming stage was distilled in the one repetition due to the low amount of raw material. That is explained in Figure 2. Also, we corrected this information in section 4.2.
Line 373 – why t-test only?
Author’s response. Not only t-test. We added the new information in section 4.3. Statistical data analysis
In conclusions – please add something about essential oil chemical components
Author’s response. Yes, we added some information.
References should be updated with some more recent ones – latest is from 2018, but a lot of them are much older.
Author’s response. Yes, we have replaced several literature sources with much more recent ones.
Reviewer 3 Report
Manuscript entitled "Variations in Yield, Essential oil and Salicylates of Filipendula ulmaria Inflorescences from Different Natural Habitats" submitted to Plants journal is well written and the results are presented in a logical and coherent manner.
The paper is adequately organized and the topic is interesting and focuses on to understand the relationship between the yield of F. ulmaria inflorescences and the qualitative and quantitative composition of the essential oil during the flowering stage in different habitats, in order to optimize collection of F. ulmaria - of valuable pharmacological raw material.
Although the manuscript is well-edited, however small improvements should be introduced that will improve its quality:
Generally to identify the phenological development stages of plants the BBCH-scale is used. Especially in scientific reports. (For example: Meier, U. (2001). Growth stages of mono- and dicotyledonous plants". BBCH Monograph. doi:10.5073/bbch0515) Therefore, I suggest using the appropriate professional numerical designations instead of (next to) the terms early and late blooming.
Line 69: the European Pharmacopoeia (2020) should be listed in the references and numbered in the text
Lines 312-321 - It is not specified when the harvest was made - both the dates in the growing season and the year. It would be worth adding, if possible, weather conditions in a given season (temperature and precipitation)
Author Response
Manuscript entitled "Variations in Yield, Essential oil and Salicylates of Filipendula ulmaria Inflorescences from Different Natural Habitats" submitted to Plants journal is well written and the results are presented in a logical and coherent manner.
The paper is adequately organized and the topic is interesting and focuses on to understand the relationship between the yield of F. ulmaria inflorescences and the qualitative and quantitative composition of the essential oil during the flowering stage in different habitats, in order to optimize collection of F. ulmaria - of valuable pharmacological raw material.
Although the manuscript is well-edited, however small improvements should be introduced that will improve its quality:
Generally to identify the phenological development stages of plants the BBCH-scale is used. Especially in scientific reports. (For example: Meier, U. (2001). Growth stages of mono- and dicotyledonous plants". BBCH Monograph. doi:10.5073/bbch0515) Therefore, I suggest using the appropriate professional numerical designations instead of (next to) the terms early and late blooming.
Author’s response. Yes, we identified the phenological development stages of plants using BBCH-scale. These stages are specified in the last paragraph of Section 4.1. Habitats and plant material.
Line 69: the European Pharmacopoeia (2020) should be listed in the references and numbered in the text
Author’s response. Yes, we corrected that in the Reference list.
Lines 312-321 - It is not specified when the harvest was made – both the dates in the growing season and the year. It would be worth adding, if possible, weather conditions in a given season (temperature and precipitation)
Author’s response. Yes, we clarified harvest dates in Chapter 4.1. Habitats and plant material. We did not indicate weather conditions in the given season for the habitats. We assume that the data of the meteorological stations, closest to the investigated habitats, could not specify the microclimatic conditions of these habitats. However, when planning this study, we took into account the fact that climate may affect the parameters studied in F. ulmaria. Two F. ulmaria habitats are located in the central district of Lithuania, three habitats – in the south-eastern Lithuanian highland district. These districts differ in climatic parameters, average annual temperature, amount of precipitation per year and the duration of snow cover. However, our statistical analysis showed that the habitats of the central Lithuania district did not significantly differ from the habitats of the south-eastern Lithuanian highland in terms of the amount of essential oils in both the early blooming stage and late blooming stage. It is the same for the parameters of salicylaldehyde and methyl salicylate in the essential oil of F. ulmaria inflorescences. We explained all this in the second paragraph of Section 3. Discussion.
Round 2
Reviewer 2 Report
Dear authors!
Thanks for significant improvements for article. Few comments that should be corrected before publishing.
New comments
Title of subchapter - Soil chemistry and its effect on composition of Thymus pulegioides essential oils – incorrect latin name - it is not Meadowsweet– please correct it.
In Table 1 as decimals somewhere dots, somewhere commas – please use the same in all article
In Fig.2. is added letters showing significant differences. Usually letters are used in increasing or decreasing order - letter a for highest value, b for significantly lower. In your case c is for lower value than d. Please change these letters.
Previous comments:
The aim of research should be more concentrated and summarized - not for 4 lines. Is it Ok to formulate with term – to understand?
Author’s response. Yes, we have shortened the aim of the research. “To understand” was changed to “to evaluate”.
Reviewer Round 2: in abstract please correct the aim - the same as introduction part
Line 254 – why t-test is used – there are comparison of more than two samples.
Author’s response. Yes, we corrected this.
Reviewer Round 2: in chapter t-test is mentioned. Please clarify where you used it.
Success in further research!
Author Response
Dear authors!
Thanks for significant improvements for article. Few comments that should be corrected before publishing.
New comments
Title of subchapter - Soil chemistry and its effect on composition of Thymus pulegioides essential oils – incorrect latin name - it is not Meadowsweet– please correct it.
Author‘s response. Yes, we corrected.
In Table 1 as decimals somewhere dots, somewhere commas – please use the same in all article
Author‘s response. Yes, we corrected.
In Fig.2. is added letters showing significant differences. Usually letters are used in increasing or decreasing order - letter a for highest value, b for significantly lower. In your case c is for lower value than d. Please change these letters.
Author‘s response. Yes, we corrected.